# GUIDED EXPLORATION IN DEEP REINFORCEMENT LEARNING

## ABSTRACT

This paper proposes a new method to drastically speed up deep reinforcement learning (deep RL) training for problems that have the property of *state-action permissibility* (SAP). Two types of permissibility are defined under SAP. The first type says that after an action $a_t$ is performed in a state $s_t$ and the agent reaches the new state $s_{t+1}$, the agent can decide whether the action $a_t$ is *permissible* or *not permissible* in state $s_t$. The second type says that even without performing the action $a_t$ in state $s_t$, the agent can already decide whether $a_t$ is permissible or not in $s_t$. An action is not permissible in a state if the action can never lead to an optimal solution and thus should not be tried. We incorporate the proposed SAP property into two state-of-the-art deep RL algorithms to guide their state-action exploration. Results show that the SAP guidance can markedly speed up training.

## 1 INTRODUCTION

Most existing Reinforcement Learning (RL) algorithms are generic algorithms that can be applied to any application modeled as a RL problem (Sutton & Barto, 2017). These algorithms often take a long time to train (Arulkumaran et al., 2017). But in many applications, some properties of the problems can be exploited to drastically reduce the RL training time. This paper identifies such a property, called *state-action permissibility* (SAP). This property can speed up RL training markedly for the class of RL problems with the property.

We propose two types of permissibility under SAP. The first type says that after an action $a_t$ is performed in a state $s_t$ and the agent reaches the new state $s_{t+1}$, the agent can decide whether the action $a_t$ is *permissible* or *not permissible* in state $s_t$. The second type says that even without performing the action $a_t$ in state $s_t$, the agent can already decide whether $a_t$ is permissible or not in $s_t$. An action is *not permissible* in a state if the action can never lead to an optimal solution and thus should not be tried. An action is *permissible* if it is *not known* to be non-permissible (i.e., the permissible action can still be non-permissible but it is not known). Clearly, the agent should avoid choosing non-permissible actions. Since the second type of permissibility is simple and we will see it in the experiment section, we will focus only on the first type. The first type is also intuitive because we humans often encounter situations when we regret a past action, and based on that acquired knowledge, we can avoid doing the same thing in an identical or similar situation in the future.

Let us use an example in autonomous driving (Figure 1) to illustrate the SAP property. In this example, the car needs to learn appropriate steering control actions to keep it driving within a lane (often called *lane keeping*). A and B are the lane separation lines (track edges), and C is the center line (track axis) of the lane. We use the term "*track*" and "*lane*" interchangeably in the paper. The ideal trajectory for the car to drive on is the center line. We assume that at a particular time step $t$ the car is in state $s_t$ (see Figure 1). It takes an action $a_t$, i.e., it turns the steering wheel counterclockwise for a certain degree. This action leads the car to the new state $s_{t+1}$. As we can see, $s_{t+1}$ is a worse state than $s_t$. More importantly, it is also quite clear that action $a_t$ is non-permissible in state $s_t$ as it would never lead to an optimal solution in the long run in accumulated reward and thus should not have been taken. When facing a similar situation in the future, the agent should avoid choosing $a_t$ to reduce the possibility of making repetitive mistakes.

The SAP property can be leveraged to drastically reduce the action exploration space of RL. Following the above example, we know that $a_t$ in state $s_t$ is not permissible as it moved the car away further from the center line. However, knowing this fact only after the action has been taken is not very

useful. It is more useful if the information can be used to help predict permissible and non-permissible actions in a new state so that a permissible action can be chosen in the first place. This is the goal of the proposed technique. Note that for type 2 permissibility, prediction is not needed.

It is important to emphasize here that choosing a permissible action at each state is by no means a greedy decision that counters the RL's philosophy of sacrificing the immediate reward to optimize for the accumulated reward over the long run (or many steps). Permissibility is defined in such a way that a non-permissible action cannot be an action that will lead to an optimal accumulated reward over the long run. The class of problems with the SAP property is also quite large as most robot functions involving navigation or physical movements and even many games of this nature have this property, e.g., flappy bird[1], pong game[1], cart-pole (Brockman et al., 2016), robot arm reacher (Brockman et al., 2016), etc. This is so because it is similar to us humans that in most cases our prior knowledge about the environment can tell us what movements/actions will not help us reach our goals. In general, RL learning with the SAP guidance is analogous to human learning which tries to smartly choose permissible/promising actions rather than blindly try all possibilities.

We propose to make use of previous states, their actions, and the permissibility information of the actions to build a binary predictive (or classification) model. Given the current state and a candidate action in the state, the model predicts whether the action is *permissible* or *non-permissible* in the state. We discuss how to make use of this predictor to guide the RL training in Sec. 4. A major advantage of the proposed predictive model is that it is trained concurrently with the RL model. It requires no human labeling of training data, which are obtained automatically during RL training by defining an *Action Permissibility* function and exploiting the SAP property (see Sec. 4). As the agent experiences more states and actions during RL training and gathers knowledge (labels) of action permissibility, the predictive model becomes more accurate (stabilizes after some time), which in turn provides more accurate guidance to the RL training, making it more efficient.

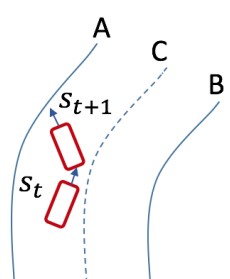

Figure 1: An illustrative example of the lane keeping task in autonomous driving.

Two questions that one may ask: (1) how to decide permissibility of an action, and (2) what happens if the predictive model predicts wrongly? For (1), the answer is that it is task/domain dependent. Our approach allows the user to provide an *Action Permissibility* (AP) function to make the decision. For (2), there are two cases. First, if a non-permissible action is predicted as permissible, it causes no issue. If a non-permissible action is chosen for a state, it just results in some waste of time. After the action is performed, the agent will detect that the action is non-permissible and it will be added to the training data for the predictive model to improve upon in the next iteration. Second, if a permissible action is predicted as non-permissible, this is a problem as in the worst case (although unlikely), RL may find no solution. We solve this problem in Section 4.

In summary, this paper makes the following contributions. **(1)** It identifies a special property SAP in a class of RL problems that can be leveraged to cut down the exploration space to markedly improve the RL training efficiency. To our knowledge, the property has not been reported before. **(2)** It proposes a novel approach to using the SAP property, i.e., building a binary predictive model to predict whether an action in a state is permissible or not ahead of time. **(3)** Experimental results show that the proposed approach can result in a huge speedup in RL training.

## 2 RELATED WORK

Exploration-exploitation trade-off (Sutton & Barto, 2017) has been a persistent problem that makes RL slow. Researchers have studied how to make RL more efficient. Kohl & Stone (2004) proposed a policy gradient RL to automatically search the set of possible parameters with the goal of finding the fastest possible quadrupedal locomotion. Dulac-Arnold et al. (2012) formulated a RL problem in supervised learning setting. Narendra et al. (2016) proposed an approach that use multiple models to enhance the speed of convergence. Among other notable works, Duan et al. (2016) proposed RL[2] to quickly learn new tasks in a few trials by encoding it in a recurrent neural network that learns through a general-purpose ("slow") RL algorithm. Wu et al. (2017) proposed a method to adaptively balance

---

[1]https://github.com/ntasfi/PyGame-Learning-Environment

the exploration-exploitation trade-off and Nair et al. (2017) tried to overcome the exploration problem in the actor-critic model DDPG Lillicrap et al. (2016) by providing demonstrations. Deisenroth & Rasmussen (2011) proposes a policy-search framework for data-efficient learning from scratch. Bacon et al. (2017) focused on learning internal policies and the termination conditions of options, and Asmuth et al. (2008) focused on potential-based shaping functions and its use in model-based learning algorithms. Although these works contribute in RL speed up, their problem set up, frameworks, and approaches differ significantly from ours. Recent advances in meta reinforcement learning have contributed in large policy improvements at test time with minimal sample complexity requirements Duan et al. (2016); Wang et al. (2016); Kulkarni et al. (2016); Finn et al. (2017), but have inadequately addressed the issue of exploration. Our work also significantly differs from these work, as we focus on guiding RL exploration by learning/leveraging the knowledge of state-action permissibilty. The recent work in (Abel et al., 2015) focused on leveraging the knowledge of action priors provided by a human expert or learned through experiences from related problems. In contrast, our work learns the state-action permissibility from the same problem. Also, Abel et al. (2015) does not introduce the concept of SAP. Several researchers also proposed some other techniques for detecting symmetry and state equivalence to speed up RL (Mahajan & Tulabandhula, 2017; Girgin et al., 2010; Bianchi et al., 2004; Osband et al., 2013; Bai & Russell, 2017). We focus on constraining the exploration space by leveraging a special property of the underlying task.

Since our approach involves prediction, it is seemingly related to model-based RL (Deisenroth & Rasmussen, 2011; Kamalapurkar et al., 2016; Berkenkamp et al., 2017; Nagabandi et al., 2018; Clavera et al., 2018), which aims to learn the transition function of the environment. However, our work is not about learning the transition probabilities and is still model-free. SAP provides a scope for encoding human knowledge into model-free setting and leverage the knowledge for fast policy learning, as opposed to learning the model of the environment in the model-based approach.

## 3 BACKGROUND

We will incorporate the SAP guidance into two deep RL algorithms, Double Deep Q Network (DDQN) (van Hasselt et al., 2016) and Deep Deterministic Policy Gradient (DDPG) (Lillicrap et al., 2016). We introduce them here, which are based on Q-learning (Watkins & Dayan, 1992). Q-learning employs the greedy policy $\mu(s) = \arg\max_a Q(s, a)$. For continuous state space, it is performed with function approximators parameterized by $\theta^Q$, optimized by minimizing the mean square loss:

$$L(\theta^Q) = \mathbb{E}_{s_t \sim \rho^\beta, a_t \sim \beta, r_i \sim \mathcal{E}}[(Q(s_t, a_t | \theta^Q) - y_t)^2]$$ (1)

where, $y_t = r(s_t, a_t) + \gamma Q(s_{t+1}, \mu(a_{t+1}) | \theta^Q)$ and $\rho^\beta$ is the discounted state transition distribution for policy $\beta$. The dependency of $y_t$ on $\theta^Q$ is typically ignored.

Recently, Mnih et al. (2015; 2013) adapted Q-learning by using deep neural networks as non-linear function approximators and a replay buffer to stabilize learning, known as Deep Q-learning or DQN. van Hasselt et al. (2016) introduced Double Deep Q-Network (DDQN) by introducing a separate target network for calculating $y_t$ to deal with the over-estimation problem in DQN.

For continuous action space problems, Q-learning is usually solved using an Actor-Critic method, e.g., Deep Deterministic Policy Gradient (DDPG) (Lillicrap et al., 2016). DDPG maintains an Actor $\mu(s)$ with parameters $\theta^\mu$, a Critic $Q(s, a)$ with parameters $\theta^Q$, and a replay buffer $\mathcal{R}$ as a set of experience tuples $(s_t, a_t, r_t, s_{t+1})$ like DQNs (Mnih et al., 2015; 2013) to store transition history for training. Training rollouts are collected with extra noise for exploration: $a_t = \mu(s) + \mathcal{N}_t$, where $\mathcal{N}_t$ is a noise process. In each training step, DDPG samples a minibatch of $N$ tuples from $\mathcal{R}$ to update the Actor and Critic networks and minimizes the following loss to update the Critic:

$$L(\theta^Q) = \frac{1}{N} \sum_i [y_i - Q(s_i, a_i | \theta^Q)^2]$$ (2)

where, $y_i = r_i + \gamma Q(s_{i+1}, \mu(s_{i+1}) | \theta^Q)$. The Actor parameters $\theta^\mu$ are updated using the sampled policy gradient:

$$\nabla_{\theta^\mu} J = \frac{1}{N} \sum_i \nabla_a [Q(s, a | \theta^Q)|_{s=s_i, a=\mu(s)} \nabla_{\theta^\mu} \mu(s | \theta^\mu)|_{s=s_i}]$$ (3)

## 4 PROPOSED TECHNIQUE

The proposed framework consists of the state-action permissibility (SAP) property, action permissibility prediction model, and the integration of the predictive model in RL to guide RL training.

### 4.1 STATE-ACTION PERMISSIBILITY

Let $r : (\mathcal{S}, \mathcal{A}) \to \mathbb{R}$ be the reward function for a given MDP with state space $\mathcal{S}$ and action space $\mathcal{A}$. In this work, we assume that the action space is one-dimensional (expressed by one variable) [2].

**Definition 1** (permissible and non-permissible action): If an action $a_t$ in a state $s_t$ cannot lead to an optimal solution (or accumulated reward) in the long run, the action is said to be a *non-permissible* action in the state. If the action $a_t$ in the state $s_t$ is not known to be non-permissible, it is *permissible*.

**Definition 2** (type 1 permissibility): Let a state transition in a RL problem be $(s_t, a_t, r_t, s_{t+1})$. We say that the RL problem has the type 1 SAP property if there is an *type 1 action permissibility* (AP1) function $f_1 : (\mathcal{S}, \mathcal{A}) \to \{0, 1\}$ that can determine whether the action $a_t$ in state $s_t$ is *permissible* $[f_1(s_t, a_t|s_{t+1}) = 1]$ or *non-permissible* $[f_1(s_t, a_t|s_{t+1}) = 0]$ in $s_t$ after the action $a_t$ has been performed and the agent has reached state $s_{t+1}$.

**Definition 3** (type 2 permissibility): Let the RL agent be in state $s_t$ at a time step $t$. We say that a RL problem has the type 2 SAP property if there is an *type 2 action permissibility* (AP2) function $f_2 : (\mathcal{S}, \mathcal{A}) \to \{0, 1\}$ that can determine whether an action $a_t$ in state $s_t$ is *permissible* $[f_2(s_t, a_t) = 1]$ or *non-permissible* $[f_2(s_t, a_t) = 0]$ without performing action $a_t$.

Clearly, a permissible action may still be non-permissible, but it is not known. Both types of action permisibility functions may not be unique for a problem. Since type 2 permissibility is simple, we focus only on type 1 permissibility. We illustrate it using an example in the lane keeping task.

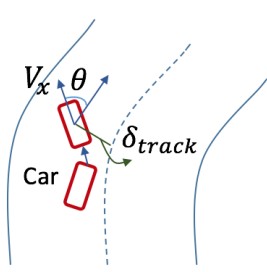

Figure 2: Visualizing the parameters of lane keeping task immediate reward function.

**Example 1.** Let, at any given time *in motion*, $\theta$ be the angle between the car's direction and direction of the track (lane) center axis, $V_x$ be the car speed along the longitudinal axis and $\delta_{track}$ be the distance of the car from the track center (center line) (see Figure 2). Given this setting, we use the following reward function (an improved version[3] of that in (Lillicrap et al., 2016)) for the lane keeping task:

$$r = V_x(\cos \theta - \sin \theta - \delta_{track}) \qquad (4)$$

From the reward function, we see that the car gets the maximum reward ($V_x$) only when it is aligned with the track axis and $\delta_{track} = 0$; otherwise the reward will be less than $V_x$.

Let $\delta_{track,t}$ and $\delta_{track,t+1}$ be the distance of the car from the lane center line (track axis) corresponding to state $s_t$ and state $s_{t+1}$ respectively. The following is an AP1 function:

$$f_1(a_t, s_t|s_{t+1}) = \begin{cases} 0 & \text{if } \delta_{track,t+1} - \delta_{track,t} > 0 \\ 1 & \text{Otherwise} \end{cases} \qquad (5)$$

This AP1 function says that any action results in the car to move further away from the track axis (center line) is not permissible.[4] This clearly satisfies the type 1 SAP property. It is type 1 because without performing the action, one will not know whether the action is permissible or not.

---

[2] We leave the multi-dimensional continuous action space case to our future work.

[3] https://yanpanlau.github.io/2016/10/11/Torcs-Keras.html

[4] Note that in a sharp turn, if the curvature of the lane is too large for the car to follow, we humans may drive to the outer side of the lane or even cut into the neighboring lane (if there is no danger) before the turn. This is usually because we drive too fast. In this work, we do not deal with speed control, which adds another dimension to our action space. We leave multi-dimensional action space RL to our future work.

### 4.2 LEARNING TYPE 1 ACTION PREMISSIBILITY (AP1) PREDICTOR

AP1 function only gives knowledge about the permissibility of an "*executed*" action. Thus, we need to continuously learn the permissibility of actions for a given state utilizing our past experiences and to predict action exploration for future states. Note that AP2 clearly does not need prediction.

As indicated earlier, AP1 prediction is a binary classification problem with two classes. Given the current state $s$ and an action $a$, the goal of the AP1 predictor is to predict whether $a$ is permissible or not permissible in $s$. Note that AP1 predictor is a learned predictive model or classifier, which is different from the AP1 function, a user provided function. The labeled training data for building AP1 predictor is produced by AP1 function $f_1$, which determines whether an action at a particular state was permissible or not permissible after the action has been performed on the state during the RL training. Each example of the training data consists of values of all variables representing a state and the action taken in the state with its class (permissible or non-permissible). After many initial steps of RL, a set of training examples for building the AP1 predictor is collected.

Since the training of the AP1 predictor is performed continuously along with the RL training, to manage the process and the stream of new training examples, we maintain a training data buffer $\mathcal{K}$ similar to the replay buffer $\mathcal{R}$ in (Lillicrap et al., 2016; Mnih et al., 2013) to train the AP1 predictor. Given a RL experience tuple $(s_t, a_t, r_t, s_{t+1})$ at time step $t$, we extract the tuple $(s_t, a_t, l(a_t))$ and store it in $\mathcal{K}$. Here, $l(a_t)$ is the class label for $a_t$ in $s_t$, permissible (+ve class) or non-permissible (-ve class) and is inferred using the AP1 function $f_1$. Similar to the replay buffer $\mathcal{R}$, $\mathcal{K}$ is finite in size and when it gets full, newer tuples replace oldest ones having the same class label $l(a_t)$.

We train AP1 predictor $E$ with a balanced dataset at a time step $t$ as follows. For time step $t$, if both the number of +ve as well as -ve tuples (or examples) in $\mathcal{K}$ are at least $N_E/2$ (ensures $N_E/2$ +ve and $N_E/2$ -ve examples can be sampled from $\mathcal{K}$), we sample a balanced dataset $\mathcal{D}_E$ of size $N_E$ from $\mathcal{K}$. Then, we train the neural network AP1 predictor $E$ with parameter $\theta^E$ using $\mathcal{D}_E$. Note that, AP1 predictor is just a supervised learning model. We discuss the network architecture of AP1 predictor used for our experiments in Appendix Section. For training $E$, we use mini-batch gradient decent to update $\theta^E$ and minimize L2-regularized binary cross-entropy loss:

$$L(\theta^E) = -\frac{1}{N_E} \sum_{(s_i, a_i, l(a_i)) \in \mathcal{D}_E} [\, l(a_i) \, log \, E(s_i, a_i | \theta^E) + (1 - l(a_i)) \, log \, (1 - E(s_i, a_i | \theta^E))] + \frac{\lambda}{2} \sum \|\theta^E\|_2^2$$

(6)

where $\lambda$ is the regularization parameter. We discuss the use of the AP1 predictor in a RL model below.

### 4.3 GUIDING RL MODEL WITH AP1 PREDICTOR

The proposed AP1 predictor can work with various RL models. In this work, we incorporate it into two deep RL models: the actor-critic model Deep Deterministic Policy Gradient (DDPG) (Lillicrap et al., 2016) and the Double Deep Q Network (DDQN) (van Hasselt et al., 2016) (see Section 3). We chose DDPG because it is a state-of-the-art for learning continuous control tasks and DDQN because it is a state-of-the-art for solving continuous state and discrete action space RL problems. Our integrated algorithm of DDPG and AP1 predictor $E$ is called DDPG-AP1, and of DDQN and $E$ DDQN-AP1. The training process of Actor $\mu$ and Critic $Q$ of DDPG-AP1 (and that of $Q$ network for DDQN-AP1) is identical to the DDPG (DDQN) algorithm (see Section 3) **except** one major modification (discussed later). The training of AP1 predictor $E$ of DDPG-AP1/DDQN-AP1 is performed simultaneously with the training of the corresponding RL model. In the following, we discuss how a trained $E$ (say, up to time step $t$) helps in guided action exploration.

Algorithm 1 presents the action selection process of DDPG-AP1/DDQN-AP1. Given the trained AP1 predictor $E$ at time step $t$, action selection for $s_t$ works as follows: Initially, DDPG-AP1/DDQN-AP1 selects action $a_t$ randomly from action space $\mathcal{A}$ via an exploration process upto $t \leq t_e$ steps (or state transitions). This phase is usually called the *Exploration Phase* in the RL literature (line 2-4). After $t > t_e$, the exploration process is not used further. For $t > t_o$, AP1 guided action selection process (line 6-18) is enabled, where $t_o < t_e$. Initially, when the RL agent starts learning, the tuples stored in $\mathcal{K}$ are few in number and thus, are not enough to build a good AP1 predictor. Moreover, $E$ also needs a diverse set of training examples (tuples) to learn well. Thus, for the initial set of steps ($t \leq t_o$), AP1 guidance is not used, but $E$ is trained. This phase is called the *Observation Phase*, which is the initial

---

**Algorithm 1** AP Guided Action Selection for RL

---

**Input:** Current state $s_t$; $\mu(s|\theta^\mu)$ as RL Action selection model (e.g, Actor network in DDPG or DDQN); AP1 predictor $E(s, a|\theta^E)$; current time step $t$; Observation time step threshold $t_o$; Exploration time step threshold $t_e$ $(> t_o)$; probability threshold $\alpha_e$ and $\alpha_{tr}$ $(> \alpha_e)$ for consulting $E$ and $v_{acc}^{t-1}(E)$ as validation accuracy of $E$ computed at time step $t - 1$.
**Output:** $a_t$: action selected for execution in $s_t$

---

1: Select action $a_t = \mu(s_t|\theta^\mu)$ for $s_t$
2: **if** $t \leq t_e$ **then**                                                    ▷ Exploration phase
3:      $a_t$ = Exploration($a_t$)                ▷ Use Noise process for DDPG and exploration strategy for DDQN
4: **end if**
5: **if** $t > t_o$ **then**                            ▷ Start AP1 guidance when $t > t_o$ (observation phase is over)
6:      **if** $t > t_e$ and $v_{acc}^{t-1}(E) \geq \delta_{acc}$ **then**
7:          Set $\alpha = \alpha_{tr}$                             ▷ $\alpha$ for training/learning phase
8:      **else**
9:          Set $\alpha = \alpha_e$                               ▷ $\alpha$ for exploration phase
10:      **end if**
11:      $l(\hat{a}_t) \leftarrow E(s_t, a_t|\theta^E)$                  ▷ Predict permissibility class label of action $a_t$ using $E$
12:      **if** $l(\hat{a}_t)$ is -ve (non-permissible) and $Uniform(0, 1) < \alpha$ **then**
13:          Select Candidate Action Space $\mathcal{A}_{s_t}$ from $\mathcal{A}$ and build $D_{s_t}$ as $\{(s_t, a) \mid a \in \mathcal{A}_{s_t}\}$    ▷ For DDPG, we sample $\mathcal{A}_{s_t}$ using low-variance uniform sampling from $\mathcal{A}$ and for DDQN, $\mathcal{A}$ being finite, $\mathcal{A}_{s_t} = \mathcal{A} - \{a_t\}$
14:          $\mathcal{A}_P(s_t) = \{a \mid E(s_t, a)$ is +ve, $(s_t, a) \in D_{s_t}\}$
15:          **if** $\mathcal{A}_P(s_t) \neq \emptyset$ **then**
16:              Randomly sample $a_t$ from $\mathcal{A}_P(s_t)$          ▷ $a_t$ is sampled from predicted permissible action space
17:          **end if**
18:      **end if**
19: **end if**
20: Return $a_t$

---

$t_o$ steps of the exploration phase. After observation phase is over $(t > t_o)$, for $t_o < t \leq t_e$ time steps, AP1 Predictor based guidance (Line 5-19) and exploration process (Line 2-4) work together, and for $t > t_e$, only AP1 based guidance (Line 5-19) works. We call this phase the *Learning/Training Phase*, where no more action exploration is done using the exploration process (line 2-4).

For $t > t_o$, the AP1 based guidance (lines 5-19) works as follows: the action selected in line 1-4 is fed to $E$ for AP1 prediction with probability $\alpha$. Some explanation is in order here about $\alpha$. $\alpha \in [0, 1)$ controls the degree by which DDPG-AP1/DDQN-AP1 consults $E$. As mentioned in Section 1, since AP1 predictor is hard to be 100% accurate, we need to deal with the case where a permissible action is predicted as non-permissible (false negative) (false positive is not an issue, see Section 1). This is a problem because in the worst case (although unlikely), RL may not find a solution. We deal with the problem by letting the Actor to listen to $E$ for $\alpha$% of the time. This probability ensures that the RL model executes the action generated by itself (including some false negatives) on environment $(1 - \alpha)$% of the time. Moreover, setting appropriate $\alpha$ also allows the RL model to experience some bad (non-permissible actions) experiences through out its training, which stabilizes RL learning. For $t_o < t \leq t_e$, we set $\alpha$ to a small value $\alpha_e$ to encourage more random exploration. Once the exploration phase is over $(t > t_e)$, $\alpha$ set to a bigger value $\alpha_{tr}$ $(> \alpha_e)$ so that DDPG-AP1/DDQN-AP1 often consults $E$ for its guidance.

In line 11, if $a_t$ is predicted as permissible by $E$, we skip lines 12-19 and action $a_t$ (selected in line 1-4) gets returned (and executed on the environment, not in the algorithm). Otherwise, in line 13, the RL model selects a candidate action set $\mathcal{A}_{s_t}$ for $s_t$ of size $N$ from the action space $\mathcal{A}$ and finds a permissible action for the current state $s_t$. For DDPG-AP1, the action space being continuous, we estimate permissible action space as follows: We sample an action set $\mathcal{A}_{s_t}$ for $s_t$ of size $N$ from the full action space $\mathcal{A}$ using low-variance uniform sampling. In this process, first, $\mathcal{A}$ is split into $N$ equal sized intervals and an action is sampled from each interval following uniform distribution to produce a set of sampled actions for state $s_t$, denoted by $\mathcal{A}_{s_t}$. Such a sampling procedure ensures that the actions are sampled uniformly over $\mathcal{A}$ with variances between consecutive samples being low. Thus, any action in $\mathcal{A}$ will be equally likely to be selected, *provided* it is predicted to be permissible (+ve) by $E$ (line 14)[5]. For DDQN-AP1, the action space being finite, we set $\mathcal{A}_{s_t} = \mathcal{A} - \{a_t\}$. Once $\mathcal{A}_{s_t}$ is selected, RL model forms a dataset $D_{s_t}$ by pairing $s_t$ with each $a \in \mathcal{A}_{s_t}$ and feeds $D_{s_t}$ to $E$ in a single batch to estimate a permissible action space for $s_t$ as $\mathcal{A}_P(s_t)$ (line 16). Here +ve means permissible. Next, RL model randomly samples an action $a_t$ from $\mathcal{A}_P(s_t)$ (line 16) and executes it on environment (not in Algorithm 1). If $\mathcal{A}_P(s_t) = \emptyset$, the original $a_t$ (selected in line 1-4) is returned

---

[5]As AP predictor does not learn the value function, it is more logical to estimate the action permissibility space and let RL find the best policy from that space. Thus, we use uniform sampling to choose action from the estimated permissible action space rather than choosing the best action (greedily) based on AP prediction score.

and gets executed. The values of the hyper-parameters $t_o$, $t_e$, $\alpha_e$, $\alpha_{tr}$ and $\delta_{acc}$ are chosen empirically (reported and discussed in Appendix).

**Modified Training of DDPG-AP1/DDQN-AP1.** Once $a_t$ gets executed on environment and the RL model receives an experience tuple $exp_t=(s_t, a_t, r_t, s_{t+1})$, we label the $exp_t$ as permissible or non-permissible using the user-provided action permissibility function. We split the Replay buffer into two equal parts, one half to store the non-permissible experiences and the other half to store the permissible experiences. When the buffer gets full, only a permissible experience can replace another permissible experience and an non-permissible experience can replace an non-permissible one.

At each step of RL training, we sample batch-size/2 experiences from the permissible section of the buffer and sample batch-size/2 experiences from the non-permissible section and then, use those samples for RL model training. This ensures a balanced training process where the RL model always gets trained on good and bad experiences, and also deals with catastrophic forgetting. Note that, in our permissibility based guidance, RL model observes more bad experiences during exploration phase compared to that in learning phase. Storing good and bad experiences in two half ensures the bad experiences do not get erased from buffer by good ones observed due to the guidance mechanism.

# 5 EXPERIMENTAL EVALUATION

We evaluate the proposed DDPG-AP and DDQN-AP techniques in the applications of the lane keeping (steering control) task and the Flappy Bird game respectively and analyze their learning performances, and compare them with the baseline.

## 5.1 LANE KEEPING TASK

We use an open-source, standard autonomous driving simulator TORCS (Loiacono et al., 2013) following (Sallab et al., 2016; 2017) for both learning and evaluation. We used five sensor readings to represent the state vector which we found are sufficient for learning good policies in diverse driving situations. The goal of our experiment is to assess how well the driving agent has learned to drive to position itself on the track/lane axis (the lane keeping task).[6] Thus, our model and baselines focus on predicting the right steering angle that can keep the car aligned with the track axis while driving with a default speed. During training, whenever the car goes out of the track, we terminate the current episode and initiate a new one. We use five diverse road tracks in our experiments. Among these 5 road tracks, we used the wheel-2 track for training and the rest of the four tracks for testing. Due to various curvature variations, we consider wheel-2 as ideal for training all possible scenarios. We present a summary of state sensors, road tracks and also, discuss network architecture and hyper parameter settings in the Appendix.

**Compared Algorithms.** Our goal is to compare our DDPG-AP (AP1 and AP2) models below with the original **DDPG** algorithm (the baseline, without any action selection guidance). Note that here we also propose a type-2 AP guidance based on some characteristics of driving.

**DDPG-AP1.** DDPG-AP1 is an extension of DDPG that uses type 1 AP function as proposed in equation 5 for the lane keeping task. Here, we use AP1 predictor for guidance.

**DDPG-AP2.** DDPG-AP2 is an extension of DDPG that applies the following two type 2 AP functions (or constraints): (1) If the car is on the left of lane center line and current action $a_t > a_{t-1}$ (previous action), instead of applying $a_t$, it samples actions uniformly from (-1.0, $a_{t-1}$)[7]. In other words, when the car is on left of the center line, it should avoid taking any left turn further. Similarly, (2) if the car is on the right of the lane center line and $a_t < a_{t-1}$, then sample actions from ($a_{t-1}$, 1.0) and it should avoid turning right further. Otherwise, the car executes $a_t$. These constraints are applied only when $\delta_{track,t} - \delta_{track,t-1} > 0$, i.e., only when the car moves away from the track center due to its previous action. If the car is moving closer to the track center, it is permissible. This method gives very strong constraints on car's movement. Clearly, this model does not need AP prediction.

---

[6]By no means are we solving the whole self-driving problem, which is much more complex. For example, in real driving, we sometimes have to go out of lane/track to avoid a collision or to be able make a shape turn due to high speed. In such a case in self-driving, the system can dynamically generate a virtual lane for the car to travel based on the current situation. We leave this, speed control, and other issues to our future work.

[7]Steering value -1 and +1 means full right and left respectively

Table 1: Performance of DDPG and DDPG-AP variants on different test tracks.

| | DDPG | | DDPG-AP2 | | DDPG-AP1 | | DDPG-(AP1+AP2) | |
|---|---|---|---|---|---|---|---|---|
| Training track: Wheel-2 [After 3k steps training] | | | | | | | | |
| Test track | Lap ? | Total Reward | Lap ? | Total reward | Lap ? | Total reward | Lap ? | Total reward |
| E-road | N (17.59%) | 4460.96 | Y | 53371.60 | Y | 53189.98 | Y | 54667.40 |
| Spring | N (4.30%) | 8258.29 | N (44.28%) | 162875.81 | Y | 368724.99 | Y | 371795.67 |
| CG Track 3 | N (5.75%) | 1574.31 | Y | 46948.97 | Y | 46364.52 | Y | 48199.51 |
| Oleth Ross | N (16.82%) | 8784.68 | Y | 105419.94 | Y | 103557.34 | Y | 107168.99 |
| | DDPG | | DDPG-AP2 | | DDPG-AP1 | | DDPG-(AP1+AP2) | |
| Training track: Wheel-2 [After 15k training steps] | | | | | | | | |
| Test track | Lap ? | Total reward | Lap ? | Total reward | Lap ? | Total reward | Lap ? | Total reward |
| E-road | Y | 40785.05 | Y | 53653.04 | Y | 56217.72 | Y | 56333.61 |
| Spring | N (36.68%) | 117519.69 | Y | 368559.39 | Y | 382746.63 | Y | 383541.04 |
| CG Track 3 | Y | 37011.54 | Y | 46975.31 | Y | 49085.96 | Y | 49535.38 |
| Oleth Ross | Y | 86275.80 | Y | 105584.83 | Y | 109506.41 | Y | 110384.84 |

**DDPG-(AP1+AP2).** Version of DDPG where we combine our DDPG-AP1 (type 1 AP) and DDPG-AP2 (type 2 AP) to give us DDPG-(AP1+AP2). Here, we learn AP1 predictor for training DDPG-(AP1+AP2) due to the use of type 1 permissibilty.

**Results and Analysis.** Figure 3 shows the comparative result of DDPG-AP variants and DDPG with regard to the average reward over the training steps. We conducted training for 15k steps and report the moving average of reward over the past 100 steps. The minor fluctuations in the curve shows the stability in learning, i.e., how smoothly each algorithm has learned to keep the car aligned to the track center axis/line. A sharp fall indicates a sudden end of episode, i.e., when the car goes out of track with a large -ve reward. We can see that the moving average reward for DDPG-AP1 and DDPG-(AP1+AP2) increases very rapidly compared to other algorithms and gets stable more quickly (around 2500 steps), whereas the learning of DDPG is quite unstable. We discuss more about the learning curves for all algorithms in Appendix.

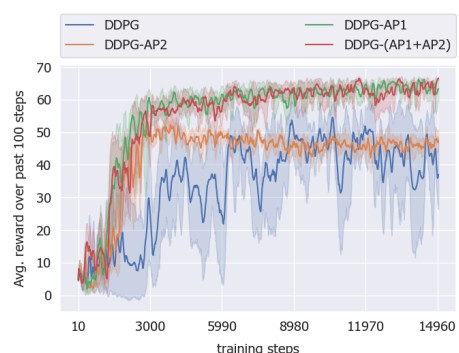

Figure 3: Avg. reward over past 100 training steps of DDPG and DDPG-AP variants on lane keeping task.

We also evaluated AP1 predictor's validation accuracy and found that the accuracy always stays above 70% during initial training steps and stabilizes with an average of 80% (see Appendix), signifying that our AP1 predictor learns well to classify permissible actions from the non-permissible ones.

Table 1 shows the performance of the algorithms on unseen test tracks considering both 3k and 15k steps of training. We use each algorithm to drive the car for one lap of each track and report the total reward obtained by each algorithm. the "Lap ?" column indicates whether the car has completed the lap or not, and if not, (%) of the total track length the car has covered from its beginning position, before it went out of track.

Considering the results for the 3k training steps (which is very few for learning a stable policy), we see that DDPG and DDPG-AP2 has not learned to make the car complete one lap for all test tracks. Both DDPG-AP1 and DDPG-(AP1+AP2) perform much better in term of lap completion. Considering 15k training steps, we see that all algorithms except DDPG have learned to keep the car on track for all test tracks. The highest total reward values and lap completion information in DDPG-(AP1+AP2) (considering all test tracks) indicate that DDPG-(AP1+AP2) has learned to find the most general policy quickly compared to others. Although DDPG-AP2 was competitive with DDPG-AP1 in 3k training steps, the rewards obtained in 15k are less than those for DDPG-AP1 and DDPG-(AP1+AP2). This shows that the policy learned by DDPG-AP2 is sub-optimal.

## 5.2 FLAPPY BIRD

Since this is a discrete action space problem, we use the RL network DDQN (see Section 3). DDQN-AP variants and DDQN network architectures and hyper parameter settings are provided in the

Appendix. We use the open source pygame version of Flappy Bird[8] for evaluation. The goal here is to make a bird learn to fly and navigate through gaps between pipes (see Figure 4(b) in Appendix), where the allowed actions are {*flap*, *no flap*}. The *flap* action causes an increase in upward acceleration and *not flap* makes the bird fall downward due to gravity.

**AP Functions.** Analyzing the Flappy bird game setting, we observed that whenever the bird flaps, it accelerates upward by 9 pixels and if it does not flap it accelerates downward by 1 pixel. An optimal solution for the game is when the expected trajectory of the bird follows the midway of the pipe gap. To achieve this and make each move safer (less prone to crashing the pipe), the bird should accelerate downward by some steps before the next flap. Thus, if the bird is above the next pipe gap center line, a flap increases the chance of hitting the upper pipe compared to that when below the gap center line. Also, if the bird is below the top surface of next lower pipe, not flapping causes the bird to fall down and reduces the possibility of reaching to the next pipe gap on the next flap without hitting the lower pipe. Based on this observation, a type-2 AP function for the game can be formulated as follows:

Let $c$ be the horizontal line that passes through the mid point of the next pipe gap and $\delta_c^t$ be the vertical distance of the agent (bird) from $c$ at state $s_t$. If $\delta_c^t > 0$, the bird lies above the gap center line $c$ and vice versa. Also, let $l$ be the horizontal line that passes through the next lower pipe Y coordinate (i.e., Y-coordinate of next gap's bottom left point) and $\delta_l^t$ be the vertical distance of the agent (bird) from $l$ at state $s_t$. If $\delta_l^t > 0$, the bird lies above the lower pipe top surface line $l$ and vice versa. Then a type-2 AP function can be defined as:

$$f_2(a_t, s_t) = \begin{cases} 0 & \text{if } C_1 \text{ or } C_2 \\ 1 & \text{Otherwise} \end{cases} \tag{7}$$

where $C_1$={$\delta_c^t > 0$, $a_t = \text{``}flap\text{''}$} and $C_2$ = {$\delta_l^t < 0$, $a_t = \text{``}noflap\text{''}$}. $C_1$ says that when the bird is above $c$, performing action "*flap*" that increases vertical acceleration (causing the bird move further up) is non-permissible. Similarly, when the bird is below $l$, performing "*no flap*" results in the bird to move further down and so is non-permissible ($C_2$). Here whether an action $a_t$ satisfies any condition in {$C_1$, $C_2$} can be determined at $s_t$. Thus, $f_2$ in equation 7 indicates type 2 permissibility.

However, even if the bird is above $l$, repeated "**no flap**" action can cause the bird to hit surface of lower pipe specially when the bird is within the pipe gap. And only when the bird crashes at $s_{t+1}$, we can conclude that "*no flap*" in $s_t$ was non permissible. Thus, we introduce a type 1 AP function (see Section 4.1) as follows:

$$f_1(a_t, s_t | s_{t+1}) = \begin{cases} 0 & \text{if } C_3 \\ 1 & \text{Otherwise} \end{cases} \tag{8}$$

where $C_3$={$\delta_l^t > 0$, $a_t = \text{``}noflap\text{''}$, $s_{t+1} = crash$}. $C_3$ indicates whether the bird has crashed to lower pipe top surface in state $s_{t+1}$ due to "*no flap*" in $s_t$. Thus, a new and stronger AP function (AP1+AP2) can be designed that combines $f_1$ and $f_2$ involving $C_1$, $C_2$ and $C_3$ which covers all our action non permissibilty cases for the Flappy bird game.

**Compared Algorithms.** We compare DDQN-AP (AP1 and AP2) models below with the original **DDQN** algorithm (the baseline, without any action selection guidance).

**DDQN-AP1.** DDQN-AP1 is an extension of DDQN that uses only $f_1$ (equation 8). Permissibility guidance for this version is very week as non-permissible experiences are only accumulated in the buffer when the bird crashes the lower pipe top surface, which are not many. Thus, this version does not result in significant improvement in speedup. Its results are not included in Table 2.

**DDQN-AP2.** DDQN-AP2 is an extension of DDQN that applies the type 2 AP function $f_2$ (equation 7). This is a much stronger DDQN-AP variant compared to DDQN-AP1.

**DDQN-(AP1+AP2).** Version of DDQN where we combine our DDQN-AP1 (type 1 AP, $f_1$ in equation 8) and DDQN-AP2 (type 2 AP, $f_2$ in equation 7) to give DDPG-(AP1+AP2). Here, we learn the AP predictor to guide the learning of RL model, and use combined AP function involving conditions $C_1$, $C_2$ and $C_3$ to label the permissibility of an action taken.

**Experimental Results.** We trained DDQN-AP2, DDQN-(AP1+AP2) and DDQN for 200k steps with $\epsilon$-greedy strategy. We conducted two experiments, In the first experiment, we used 30k exploration time steps and in the second experiment, we used 60k exploration time steps. 1000 initial observation

---

[8]github.com/yenchenlin/DeepLearningFlappyBird

Table 2: Average test scores over 50 games (episodes) of DDQN, DDQN-AP2 and DDQN-(AP1+AP2) on hard difficulty level (pipe gap = 100) of the Flappy bird game. The 50 games consists of five test experiments conducted with five different random seeds (10 games have been played in each test experiment by the model trained on a given random seed).

| training steps | exploration steps = 30k | | | exploration steps = 60k | | |
|---|---|---|---|---|---|---|
| | DDQN | DDQN-AP2 | DDQN-(AP1+AP2) | DDQN | DDQN-AP2 | DDQN-(AP1+AP2) |
| 100k | 20.96 | 55.22 | 165.4 | 1.68 | 49.98 | 44.48 |
| 150k | 49.5 | 94.24 | 349.04 | 25.88 | 108.5 | 318.04 |
| 200k | 80.62 | 148.8 | 663.74 | 77.14 | 181.06 | 827.42 |

steps were used in both cases. In both experiments, we observed drastic growth in reward during training for both DDQN-AP2 and DDQN-(AP1+AP2) compared to that for DDQN. The learning curves of DDQN and DDQN-AP variants for the 60k exploration experiments are shown and discussed in Appendix. We also noted that AP predictor's validation accuracy always stays above 90% during training and stabilizes with an average of 97.8% (discussed more in Appendix).

Next, we evaluate the performance of the trained DDQN-AP2, DDQN-(AP1+AP2) and DDQN in terms of the average test score achieved by each algorithm over 50 test games (episodes). Table 2 shows the test performance of the said algorithms recorded after 100k, 150k and 200k training steps. We see that DDQN-AP2 performs significantly better than DDQN (baseline) and DDPG-(AP1+AP2) outperforms them both by a large margin. Note that the average scores for all three algorithms for 100k and 150k training steps are higher for the 30k exploration steps experiment compared to the 60k exploration steps one. This is because with only 30k exploration steps, the algorithms gets a longer training (post-exploration) phase for the network than with 60k exploration steps. For example, considering the 150k training steps evaluation, the learning phase has 120k steps for the 30k exploration steps experiment but only 90k steps for the 60k exploration steps one. However, at the 200k training step, DDQN-AP2, DDQN-(AP1+AP2) in the 60k exploration steps experiment performs significantly better than that for the 30k exploration steps one as the sufficiently longer exploration phase introduces more stability in learning and assists AP based guidance in accelerating the learning process in post-exploration/training phase.

## 6    DISCUSSION AND FUTURE WORK

This work deals with a class of RL problems with the SAP property. Examples of this class of problems *primarily* include robot navigation problems, planning for solving a task, some games, etc. For such a problem, it is often not hard to specify an AP function with expert's domain knowledge. However, by no means do we claim that the SAP property is applicable to all RL problems. Some RL problems don't have the SAP property or it is hard to specify AP functions for them (e.g., environments with high dimensional action space like humanoid Brockman et al. (2016)). We should note that this work does not focus on designing procedures to identify AP functions as AP functions depend on specific application and expert's knowledge. Also, it does not require the user to provide an "*optimal AP function*" as several AP functions can usually be designed for a given problem. This work mainly aims to provide a framework to guide the existing Deep RL algorithms when a fairly good AP function can be designed for an environment. In the future, we plan to extend our framework to multi-dimensional continuous and discrete action spaces and apply it to other practical applications.

## 7    CONCLUSION

This paper proposes an novel property, called state-action permissibilty (SAP), for improving the RL training efficiency in problems with this property. To leverage this property, we proposed two types of action permissibility (AP1 and AP2) to help DRL algorithms select promising actions to speed up training. This is analogous to human RL learning in which we always smartly choose permissible actions based on our prior knowledge rather than randomly/blindly try all possibilities. Our experiments showed that the proposed method is highly effective.

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

APPENDIX

THE TORCS SIMULATOR

The Open Racing Car Simulator (TORCS) provides us with graphics and physics engines for Simulated Car Racing (SCR). The availability of the diverse set of road tracks with varying curvatures, landscapes and slopes in TORCS makes it an appropriate choice for model evaluation in different driving scenarios. It also allows us to play with different car control parameters like steering angle, velocity, acceleration, brakes, etc. More details can be found at `https://www.cs.bgu.ac.il/~yakobis/files/patch_manual.pdf`. For our lane keeping (steering control) problem setting, we identified and used five sensor variables that are sufficient for learning the steering control action as presented in Table 3. The *trackPos* parameter in Table 3 has been used as a parameter for

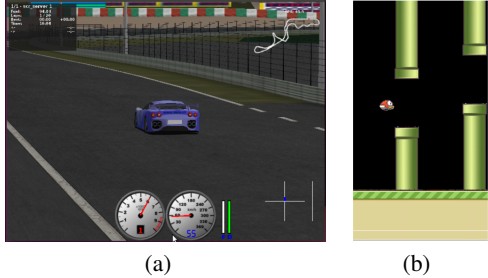

(a)               (b)

Figure 4: Snapshot of (a) Torcs simulator (b) Flappy Bird game.

designing the AP1 function in our concerned lane keeping task (see Equation 5 in the paper). Figure 4(a) shows a snapshot of TORCS simulator.

Figure 5 shows the five tracks (along with their lengths) used for evaluation. These tracks are diverse in landscapes and slopes. The Wheel-2 track is used for training the model and the rest four tracks are used for testing.

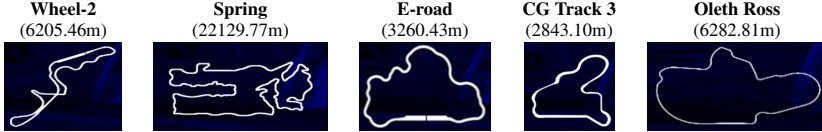

Figure 5: Various road tracks (with track length) used in our experiments.

LANE KEEPING: RL MODEL IMPLEMENTATION DETAILS

**Network Architecture.** For our lane keeping (steering control) task, the Actor network is a feed forward (fully connected) network with 128 units in layer-1 and 256 units in layer-2 followed by the action projection (output) layer. In the Critic network, we first learn state representation $s$ using two fully connected layers of 128 and 256 units. We also learn a representation of the action $a$ chosen by Actor at state $s$ with one fully connected layer of 256 units. Then, we concatenate $s$ and $a$ and learn a combined representation with a fully connected layer of 256 units before projecting it into Q-value (the output layer) for the state $s$ and given the action $a$ in $s$. This implementation of the Actor and Critic networks is inspired by a related open-source implementations available on Web[9].

The network architecture for AP1 predictor is identical to that of Critic except that instead of Q-value, the combined representation of $s$ and $a$ is projected into two class (binary classification) output through a softmax projection (classification in this case) layer. We train both networks with Adam optimizer.

**Hyper-parameter Settings.** The important empirically chosen parameters of the model are: learning rates for Actor is set as 0.0001, Critic as 0.001 and AP predictor as 0.001, the regularization parameter $\lambda$ as 0.01, $\delta_{acc} = 0.7$, discount factor for Critic updates as 0.9, target network update parameter as 0.001, $\alpha_e$ as 0.5 and $\alpha_{tr}$ as 0.9, replay buffer size as 100k, knowledge buffer size as 10k (stores tuples in 9:1 ratio as training and validation examples), batch size as 128, sample size as 128 used for AP1-based guidance. $t_o$ is set as 200 and $t_e$ is set as 1200 for both 15k training and 3k training experiments. Sample size for building the dataset for training AP1 predictor at each step is set as 2k and validation sample dataset size as 200 which is used to compute validation accuracy of AP1 predictor at each step of RL training. We employed the popularly used Ornstein-Uhlenbeck process for noise-based exploration with $\sigma = 0.3$ and $\theta = 0.15$ following standard settings for DDPG exploration.

FLAPPY BIRD: RL MODEL IMPLEMENTATION DETAILS

Figure 4(b) shows a snapshot of the Flappy Bird game for textithard difficulty level.

---

[9]github.com/yanpanlau/DDPG-Keras-Torcs

Table 3: TORCS state and action variables along with their descriptions used in our experiments.

| Name | Range (unit) | Description |
|------|-------------|-------------|
| State Variables | | |
| angle | $[-\pi, \pi]$ (rad) | Angle between the car direction and the direction of the track center axis. |
| trackPos | $(-\infty,+\infty)$ | Distance between the car and the track center axis. The value is normalized w.r.t to the track width: it is 0 when car is on the axis, -1 when the car is on the right edge of the track and +1 when it is on the left edge of the car. Values greater than 1 or smaller than -1 mean that the car is out of track. |
| speedX | $(-\infty,+\infty)$(km/h) | Speed of the car along its longitudinal axis. |
| speedY | $(-\infty,+\infty)$(km/h) | Speed of the car along its transverse axis . |
| speedZ | $(-\infty,+\infty)$(km/h) | Speed of the car along its Z-axis |
| Action | | |
| Steering | $[-1, 1]$ | Steering value: -1 and +1 means respectively full right and left, that corresponds to an angle of 0.366519 rad. |

**Network Architecture.** We use deep convolution network for constructing the double DQN (DDQN) following (Hasselt et al. 2015) and an existing open-source implementations available on Web[10]. The input to the double DQN network is a 80x80x4 tensor containing a rescaled, and gray-scale, version of the last four frames. The first convolution layer convolves the input with 32 filters of size 8 (stride 4), the second layer has 64 filters of size 4 (stride 2), the final convolution layer has 64 filters of size 3 (stride 1). In between the first and second convolution layer, we apply a max pooling layer of size 2 (stride 2) with 'SAME' padding. The representation obtained in the third convolution layer is flattened and fed to a fully-connected (FC) hidden layer of 512 units to get a representation (say, hidden representation $h_s$) which is then projected into Q-value (output layer) of size 2 (there are two possible actions for the Flappy bird game, flapping or not flapping).

The network architecture of the AP1 predictor is built as a shared network (shared weights) with that of DDQN upto the layer learning the state representation $hfc1$. We use a FC layer of 256 units to learn representation of an action $a$. Then, we concatenate two representations (i.e., $hfc1$ and representation of $a$) and learn a combined representation of the concatenated vector using another FC layer of 256 units. Finally, the combined representation is projected into two class (binary classification) output through a softmax projection layer. We train both DDQN and AP1 predictor networks with Adam optimizer.

Note that, due to the shared representation learning of state $s$, the parameters of the shared network are trained with both AP1 predictor loss (equation 6) and RL loss (equation 2). The AP1 predictor loss being a supervised learning loss function with fixed target labels (unlike estimated target Q values) accelerates the training. For the lane keeping task, the network architecture being much simpler, we can train two networks (RL and AP1 predictor) quickly without the need for learning a shared representation of the state variable.

Although DDQN-AP2 does not require an AP predictor, we observed performance improvement when we trained DDQN-AP2 parameters with cross entropy loss [like in DDQN-(AP1+AP2)] over examples annotated by type-2 AP function apart from the RL loss. Note that, as DDQN-AP2 performs non permissible actions with (1-$\alpha$)% probability, we can use type-2 AP function to label the executed actions and populate a knowledge buffer just like in case of AP1 based guidance. The results reported in Table 2 corresponds to this version of DDQN-AP2.

**Hyper-parameter Settings.** For Flappy bird, the empirically chosen hyper-parameters are: learning rates for DDQN as 5e-6 and AP predictor as 0.0001, regularization parameter $\lambda$ as 0.01, $\delta_{acc} = 0.95$, discount factor as 0.95, target network update parameter as 0.001, $\alpha_e$ as 0.3 and $\alpha_{tr}$ as 0.8, replay buffer size as 50k, knowledge buffer size as 25k (stores tuples in 9:1 ratio as training and validation examples), batch size for training as 128, sample size as 2 used for AP-based guidance (as there are two possible actions for Flappy bird), $t_o$ as 1000, $t_e$ as 30k for 30k exploration steps training experiments (see Table 2), $t_e$ as 60k for 60k exploration steps training experiments, sample size for building dataset for training AP1 predictor at each step as 2k, and validation sample dataset size as 200 which is used to compute validation accuracy of AP1 predictor at each step of the training.

For training of DDQN and DDQN-AP, we use $\epsilon$-greedy strategy for the action space exploration. For 30k annealing steps training experiments, we set initial $\epsilon$ as 1.0, final $\epsilon$ as 0.01 and annealing

---

[10]github.com/yenchenlin/DeepLearningFlappyBird

steps as 30k with observation phase of 1k steps. For 60k annealing steps training experiments, we set annealing steps as 60k keeping all other parameters same as that for 30k.

**Accelarated Training of DDQN-AP variants for Flappy Bird.** The training of AP variants shows drastic improvement over the baseline algorithm, when the reward for a non-permissible transition is less than that for permissible one. In lane keeping task, the (continuous valued) reward function in equation 4 ensures that any non-permissible action (labeled by the AP1 function in equation 5) will always have less reward than that for a permissible one in a state.

However, unlike lane keeping, for Flappy bird, often permissible and non-permissible actions (labeled by the AP2 functions in equation 7) receives the same reward of 0.1 from the environment. This is because, in the game, whenever the bird crosses a pipe, it gets 1.0 immediate reward; if it crashes, it gets -1.0 immediate reward and otherwise, if it remains alive, it gets 0.1 always. Due to such (discrete) reward function for the game, even if the AP1 predictor and AP functions (including type 1 and type 2) can differentiate between good and bad moves during training, the RL model doesn't learn the knowledge online. Rather, it slowly figures it out using Bellman equation in a delayed learning process. This basically diminishes the advantage of using AP guidance.

To alleviate this problem, we introduced the idea of instant policy rectification, i.e., whenever the bird executes a non-permissible action $a_t$ and receives a non-permissible experience, it assumes that it has (virtually) crashed. Thus, for a non-permissible experience, the bird (virtually) ends the episode with an immediate reward of -1.0 and end of episode flag being true. Thus, for all non-permissible experiences stored in the replay buffer, we train the DDQN-AP variants with target Q value of -1.0 (i.e. the target Q value for a real experience causing crash) and for all permissible experiences in replay buffer, we follow the traditional Bellman equation and target Q network to estimate the target Q value. This drastically accelerated the training of DDQN-AP variants.

ADDITIONAL EXPERIMENTAL RESULTS AND DISCUSSIONS

**Hyper-parameter Tuning for AP-Guided Exploration.** The parameters, $t_o$ and $t_e$ are adopted from existing RL methods, which denote the number of steps in the observation phase (during which epsilon is set as 1.0) and the number of steps in the exploration phase (during which epsilon value is annealed from 1.0 to a low value, here 0.01) respectively.

The newly introduced parameters $\alpha_e$, $\alpha_{tr}$ and $\delta_{acc}$ have their distinct objectives as follows: $\alpha_e$ and $\alpha_{tr}$ are the values of alpha in exploration and post exploration (training) phase respectively and control the degree by which agent listens to AP predictor. If we set $\alpha_e = 0$, it denotes the agent does not utilize the permissibility knowledge at all during the exploration phase and so, the growth of the learning curve will be slow compared to that for $\alpha_e > 0$. Generally, we keep $0 < \alpha_e \leq 0.5$ to encourage more diverse exploration which also help in AP predictor training and quickly populates non-permissible replay buffer. $\alpha_{tr}$ is generally set $1 > \alpha_{tr} \geq 0.5$. High $\alpha_{tr}$ indicates the agent will more often listen to the AP predictor and thus, explore good actions more often than repeatedly executing bad actions (non-permissible ones).

$\delta_{acc}$ helps in measuring the validation performance of the AP predictor and has two objectives: (1) It sets an accuracy threshold on the performance of the AP predictor so that if the predictors validation accuracy is $< \delta_{acc}$, it indicates the learned model is not reliable and needs more training; (2) if validation accuracy is $\geq \delta_{acc}$, then the learned model is considered reliable to assist the agent in exploration. Hence, after training for an initial number of steps, if the validation accuracy stays $\geq \delta_{acc}$, we can postpone the training of AP predictor until its validation accuracy at a given step falls below $\delta_{acc}$. We also found that setting these hyper-parameters does not require very exhaustive fine tuning and they are robust to agent's performance (considering minor changes in their tuned values).

**Learning curves.** Figure 6 shows the average number of episodes consumed by each algorithm in 15k training steps (or state transitions) noted over 5 training experiments with various random seeds. If an algorithm consumes less number of episodes, it means the algorithm learns quicker to keep the car moving without going out of track. From Figure 6, we observe that DDPG took more than 100 episodes (on avg.) to learn to drive for a considerable amount of distance. However, the sharp falls in the average reward and initiation of a new episode indicate that the learning is yet not stable. Among all, DDPG-AP1 and DDPG-(AP1+AP2) learn very quickly, and as their curves do not fall down, which indicates the car has never gone out of track after $20^{th}$ (for DDPG-(AP1+AP2)) and

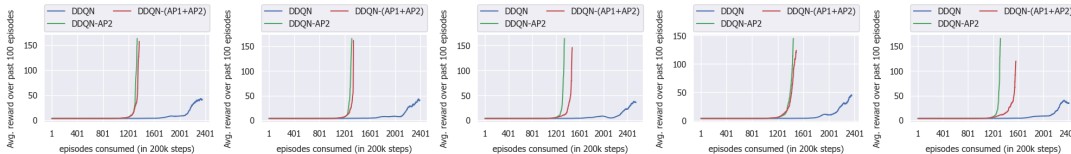

Figure 6: Avg. reward per episode for DDPG and DDPG-AP variants on lane keeping task over five training experiments with different random seeds.

Figure 7: Avg. reward over past 100 episodes for DDQN and DDQN-AP variants on Flappy bird over five training experiments (60k exploration phase) with different random seeds.

$26^{th}$ (for DDPG-AP1) episodes on average. Overall, DDPG-(AP1+AP2) not only learns quicker but also achieves best performance in terms of per episode reward and in test runs (see Table 1).

Figure 7 shows the average reward over past 100 training episodes (or games) for DDQN and DDQN-AP variants and episodes consumed by each algorithm in 200k training steps (state transitions). From Figure 7, we see that both DDQN-AP2 and DDQN-(AP1+AP2) trains much more rapidly than the baseline DDQN which is reflected in the escalating growth of the avg. reward curve and avoids collision with pipes for a longer period of time, consuming less episodes compared to DDQN. Although both DDQN-AP2 and DDQN-(AP1+AP2) learns rapidly, the learning of DDQN-(AP1+AP2) is more stable due to the incorporation of type-1 AP function (eqn. 8) in learning, covering all possible non-permissibility cases, which is reflected in its test performance (see Table 2).

**Validation Performance of AP1 Predictor.** Figure 8 shows the validation accuracy of AP1 classifier(s) for initial no. of training steps. As we can see in both 8(a) and 8(b), the validation accuracy of the AP classifiers increases over time with the incoming examples labeled by the corresponding AP function and then, gradually saturates to fairly high accuracy, denoting the learning has becomes stable. This suggests that, we

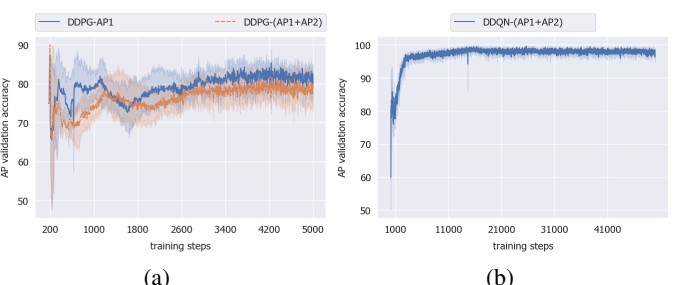

(a)                    (b)

Figure 8: Validation accuracy of AP predictor/classifier over training steps for (a) lane keeping task and (b) flappy bird (corresponding to the 60k exploration training experiment).

do not need to train the predictor for all steps during whole training period. During RL training, we postpone the training of the predictor, whenever the validation accuracy is above a threshold ($\delta_{acc}$) and resume its training whenever the validation accuracy falls below the threshold util it again goes above $\delta_{acc}$. Here, $\delta_{acc}$ is considered as the threshold for predictor's reliability.

