# OpenReview forum: "Guided Exploration in Deep Reinforcement Learning"
_ICLR.cc/2019/Conference_

### Official Review · AnonReviewer2 · 2018-11-01
**The application of SAP seems very narrow.**

**Rating:** 3
**Confidence:** 3

**Review:**

This paper proposed the concept of state-action permissibility (SAP). Given a user-defined type 1 SAP function, the algorithm learns a classifier to predict whether an action at a given state is permissible or not. Based on this prediction, the reinforcement learning (RL) algorithms can limit the exploration only to the permissible actions, and thus greatly reduce the cost of learning. The proposed algorithms are tested on two simple tasks, both of which have the same flavor of following a predefined track.

Although the results of the experiments show that SAP helps to speed up RL, I think that the application of SAC is very narrow. It is extremely difficult to define an AP1 function in general. For example, for most of the OpenAI gym environments (such as half-cheetah, ants or humanoid), it is not clear to me how to manually define an AP1 function. It would be more convincing if the paper can apply the proposed techniques to some of the benchmark OpenAI gym environments.

Even for the lane following task described in the paper, the AP1 function in eq. 5 is limited and eliminates many good solutions. It constrains that the action should not lead to more deviations to the center line in the next time step. This greedy constraint will not work in more interesting driving scenarios. For example in a sharp turn, if the curvature of the lane is too large for the car to follow, a common strategy (that can be learned by vanilla RL algorithms) is to first drive to the outer side of the lane before the turn, cut to the inner side at the turn and exit the turn to the outer side. This optimal solution to negotiate a tight turn is completely eliminated by the user-defined AP1 function (eq. 5).

The idea of AP1 is somewhat contradictory to the philosophy of reinforcement learning. AP1 is a greedy decision based on the next step while RL optimizes for the accumulated reward over many steps. RL allows taking an action that will sacrifice the immediate reward (e.g. deviate from the center line of a lane) in the next step but can accumulated more reward in the long run (successfully drive along a tight turn). In most of cases, by looking at the next state, it is just not possible to predict whether a specific action cannot lead to the optimal long-term reward (SAP).

For the above reasons, I think that the application of SAP would be very narrow, especially for reinforcement learning. I would not recommend accepting this paper at this time.

---

> ### Author Response · Authors · 2018-11-18
> **Response to AnonReviewer2**
>
> We thank you for your valuable comments. Please find our responses below.
>
> C1: Although the results of the experiments show that SAP helps to speed up RL, I think that the application of SAP is very narrow….
>
> R1: We agree that designing a good AP function for complex environments with too many parameters can be challenging, although not impossible. Also, note that, our work does not require user to provide the “optimal AP function” for a given problem. Several AP functions can be designed for a given problem (as we have discussed in the paper). We aim to provide a framework for existing Deep RL where if a fairly good AP function can be designed for an environment (often it’s not that difficult to come up with a fairly good AP functions for many environments), the proposed technique can result in drastic speed up. In other words, we show that the idea of SAP is useful. Designing a good AP function for a complex scenario requires more analysis and knowledge of the Application, but it certainly does not make the idea inapplicable. By no means do we say that for every task there is at least one AP function. As we stated, our goal here is to help speed up a class of problems.
>
> We do believe that most robot functions involving movements and navigations have the SAP property, which is not a small application domain. We can often design a fairly good AP function for robot navigation using just common sense. Thus, by learning an AP predictor, we can quickly cut-off unnecessary action exploration and speed up the learning. We believe that is what we humans do as we smartly choose permissible actions rather than blindly try everything.
>
> Most of mentioned benchmarks in gym are for multidimensional action space. In this work, we only deal with one dimensional discrete/continuous action space. We leave the multidimensional case as our future work (also mentioned in footnote 1). Hence, these environments are not suitable for our experiments.
>
> C2: Even for the lane following task described in the paper, the AP1 function in eq. 5 is limited and eliminates many good solutions…
>
> R2: As an RL problem can have multiple reward functions, it can have multiple AP functions as well, depending on the goal of the task in hand.  It’s true that our proposed AP function (eqn. 5) will not work in more complex driving scenarios but we do not aim to do that in this paper at the moment. Solving the complete autonomous driving problem is out of the scope of this work. Rather, we use a specific task (lane keeping) as our test bed to prove our hypothesis -  the idea of SAP is useful to speed up RL. Thus, we don't focus on designing a complex AP function (to cover all cases).
>
> Regarding the suggested strategy of “driving to the outer side of the lane before the turn, cut to the inner side at the turn,” we have a different opinion. We believe that is a speed control problem, which we do not study in this paper. In real life, this happens normally because we did not slow down enough at the turn which forces us to go to the outer lane. At least, that is the case for me. If the speed is also controlled by the RL, this scenario should be avoided because it is quite dangerous unless there is no car in the outer lane. Thus, in the RL learning phase, this kind of behavior should be penalized in speed control policy learning. This scenario could happen when the angle is so sharp that it is impossible to turn without cutting into the other lane (e.g., at some U-turn locations), but in that case, an autonomous car system normally will generate a new virtual lane for the car to follow (we have worked on real-life self-driving cars in the field). Another option is just to turn the steering wheel with the maximum angle possible. In both cases, the proposed RL framework still works.
>
> C3:  The idea of AP1 is somewhat contradictory to the philosophy of reinforcement learning…
>
> R3: The purpose of AP function is only to cut off exploration space for given a state, and enabling RL to not explore non-permissible actions in similar states again and again. In particular, it estimates a permissible action space in a given state and prioritize exploration of those actions in that state compared to the non-permissible ones. There may be multiple permissible actions in a given state to choose from. But, SAP does not tell you which one is optimal at that point. Rather, SAP only tells you which one you should definitely avoid exploring, as there is a better option (action) available in that state to explore. And, it’s the RL’s job to find out the optimal policy (optimal action) from the permissible action space in the long run. Thus, as we are not chopping off any optimal solution in AP-based guidance [note, even the RL agent always explores non-permissible actions with (1- alpha) probability], we believe the idea of SAP is not contradictory to RL. Similar to human driving, we do not try all possible options as we can predict what actions are definitely not good.

---

### Official Review · AnonReviewer1 · 2018-11-02
**A simple but nice idea. However, there are issues with the algorithm in the continuous action case and the evaluation could be more exhaustive.**

**Rating:** 5
**Confidence:** 4

**Review:**

The paper introduces permissible actions to reinforcement learning problems. A action is non-permissible if it is known to not lead to the optimal solution. The agent can, after executing an action a_t in state s_t and ending up in s_t+1, estimate whether the action is a_t is non-permissible. This data is used to train a new classifier that predicts the permissibility of an action in a state. The exploration of the RL algorithm can now be guided by the permissibility estimate, i.e., non-permissible actions are not executed.

The paper is well written and presents a simple, but promising idea to simplify reinforcement learning methods. I so far have not seen the definition of non-permissible actions in the literature so I believe this is novel and makes intuitively also sense, as permissible actions can be identified in many scenarios. However, the paper has a few issues that I want the authors to address:
- The amount of newly introduced hyperparameters is quite big and I am not sure whether the improved performance justifies the increased number of hyperparameters justifies.
- How many trials have been used to generate the results? Fig3 says "Avg. reward over past 100 training steps". Does that mean only one trial and you average over the last 100 rewards? In order to be significant, at least 5 to 10 trials have to be used as deep RL is known to show highly varying results depending on the random seed. Please also report error bars.
- Why are there no learning curves for Flappy Bird?
- The method for creating the action set if the selected action is permissible seems very adhoc for me, at least in the continuous action case. Would it not make more sense to include the gradient of the classifier into the actor update of DDPG such that the policy would also learn to avoid non-permissible actions? The presented method is in my opinions very hard to scale to higher dimensional action spaces (>2), which is quite a limitation of the approach.
- The description of Section 4, in particular of the construction of the candidate actions could be made more clear.
- Results are only shown for a rather low dimensional action set (driving) and a discrete action example. 1-2 more illustrations where AP1 could be useful would be highly appreciated.

---

> ### Author Response · Authors · 2018-11-25
> **Response to AnonReviewer1**
>
> We thank you for your valuable comments. Please find our response below.
>
>
> C1: The amount of newly introduced hyperparameters is quite big and I am not sure whether the improved performance justifies the increased number of hyperparameters.
>
> R1: Compared to the traditional exploration methods (e.g., epsilon-greedy exploration) in RL, we have only introduced three extra parameters: alpha_e, alpha_tr and delta_acc. The parameters, t_o and t_e are adopted from existing RL methods, which basically denote the number of steps in observation (during which epsilon is set as 1.0) and exploration (during which epsilon value is annealed from 1.0 to low value, here 0.01) phases respectively. The newly introduced parameters alpha_e, alpha_tr and delta_acc have their distinct objectives as follows: alpha_e and alpha_tr are the values of alpha in exploration and post exploration (training) phase respectively and control the degree by which agent listens to AP1 predictor. delta_acc helps in measuring the validation performance (reliability) of the AP1 predictor. We discussed more about hyper-parameter tuning strategies in Appendix of the revised version.
>
>
> C2: How many trials have been used to generate the results? Fig3 says "Avg. reward over past 100 training steps". Does that mean only one trial and you average over the last 100 rewards? In order to be significant, at least 5 to 10 trials have to be used as deep RL is known to show highly varying results depending on the random seed. Please also report error bars.
>
> R2: We have used 5 trials for each algorithm in the experiment and also reported the error curves in the revised version. Please see Section 5 and Additional Experimental Results section in Appendix.
>
>
> C3: Why are there no learning curves for Flappy Bird?
>
> R3: We have added the learning curves for Flappy Bird in the revised version (please find it in Appendix).
>
>
> C4: The method for creating the action set if the selected action is permissible seems very adhoc for me, at least in the continuous action case. Would it not make more sense to include the gradient of the classifier into the actor update of DDPG such that the policy would also learn to avoid non-permissible actions? The presented method is in my opinions very hard to scale to higher dimensional action spaces (>2), which is quite a limitation of the approach.
>
> R4: Thanks for pointing this out. In fact, we briefly mentioned about this in Appendix of our submitted version. For Flappy bird, we learn a shared network of AP1 predictor and the DDQN. Thus, the gradient update due to cross entropy loss optimization for training AP predictor also affects the learning of DDQN and helps in accelerated and stable training. We also apply similar ideas in DDQN-AP2 training, although it does not require an AP1 predictor. For steering control problem, as all DDPG-AP variants learn very quickly (see Figure 7 in Appendix) and have much less parameters to learn compared to that in Flappy bird, we did not feel the need to apply this idea to train the models, although, the idea is applicable to both cases.
>
> We believe the proposed idea of SAP can be extended to multiple action dimensions as well. For example, considering autonomous driving, we can define three AP functions independently, one for steering control (as we did in our work) and other two for speed control like for break and acceleration. Their interaction will be quite interesting. We feel it will improve RL learning even further because the reduction in each dimension will result in much more reduction in the cross product. We leave the formulation of SAP for this multidimensional action space case as our future work (as mentioned in footnote 2).
>
>
> C5: The description of Section 4, in particular of the construction of the candidate actions could be made more clear.
>
> R5: We have updated section 4.3 with a footnote on the sampling of the candidate actions. Note that, we randomly sample an action at uniform from estimated permissible action space for the agent to explore. We found it to work better than probabilistic sampling over the action space by choosing the best action (greedily) based on AP1 prediction score. As AP1 predictor does not learn the value function, it is more logical to estimate the permissibility space and let RL find the best policy from that space.
>
>
> C6: Results are only shown for a rather low dimensional action set (driving) and a discrete action example. 1-2 more illustrations where AP1 could be useful would be highly appreciated.
>
> R6: In our work, we only deal with one dimensional continuous/discrete action space and evaluated our model based on that. Our main goal in this paper is to introduce the idea of SAP and empirically show that SAP is useful for RL speed up. We leave the formulation of SAP for the multidimensional case as our future work (as mentioned in footnote 2).

---

### Official Review · AnonReviewer3 · 2018-11-05
**A constrained learning of permissable action-state space for speeding up RL**

**Rating:** 7
**Confidence:** 5

**Review:**

The authors introduce an approach for constraining the action-state space of RL algorithms, with the premise to speed up their learning. To this end, two types of constraints are introduced, coupled and embedded into the traditional policy learning for RL. The main idea of using a binary predictors for predictions of permissible actions leading to desired  states is interesting and novel. It is an intuitive approach for constraining the space and the authors showed in their experiments that it leads to significant speed up in learning of two common RL methods (DDQN and DDPG). The approach is also motivated by recent trends in meta-learning (of the binary predictor) and it would be good if the authors relate it to that (also citing some literature on meta learning).

While I am in favor of accepting this paper, I think there are several aspects that need be commented on/addressed:

- what would be a simple baseline for constraining the action-state space? One possibility could be to use the learned model to simulate the trajectories and based on that hard code the constraints? Any other ideas, task-specific?

- what is the relation to the model-based RL? In model-based RL we try to learn the transition probabilities from action to states. Could we impose any sparsity constraints on such a model to achieve a similar performance. While the proposed model is more elegant in that it allows the learning of the predictors on the fly, I feel there is a lack of comparisons with approaches that could easily be implemented using heuristics. Please comment.

- could you be more precise about how often the prediction model is updated? What are potential adverse effects if this models keeps overfitting?

There are also limitations in terms of the number of hyperparameters that need be fine-tuned. I would like that the authors include one paragraph discussing in more detail the limitations of their approach.

---

> ### Author Response · Authors · 2018-11-25
> **Response to AnonReviewer3**
>
> We thank you for your valuable comments. Please find our responses below.
>
> C1: The approach is also motivated by recent trends in meta-learning (of the binary predictor) and it would be good if the authors relate it to that (also citing some literature on meta learning).
>
> R1: We have updated Sec 2. with recent works on meta-learning for RL.
>
> C2: what would be a simple baseline for constraining the action-state space? One possibility could be to use the learned model to simulate the trajectories and based on that hard code the constraints? Any other ideas, task-specific?
>
> R2: In fact, AP2 functions are actually constraints because they prevent some non-permissible actions from being taken without learning/prediction. However, our experiments showed that AP1, which needs learning, helps significantly.
>
> C3: what is the relation to the model-based RL? In model-based RL we try to learn the transition probabilities from action to states. Could we impose any sparsity constraints on such a model to achieve a similar performance? While the proposed model is more elegant in that it allows the learning of the predictors on the fly, I feel there is a lack of comparisons with approaches that could easily be implemented using heuristics. Please comment.
>
> R3: Thanks for pointing out this connection. However, our work is not about learning the transition probabilities. It is about learning to constrain the exploration space, a binary relation, permissible or not permissible. Our work is still model-free. We discuss this in Sec. 2 of the revised version. Accurately learning the model of the environment (specially, considering continuous state space problems) is often difficult in practice. Thus, the model free approach is widely used. SAP provides a scope for encoding human knowledge into model-free setting and leverage the knowledge in fast policy learning. The motivation is – humans may not provide the optimal policy for a given state, but can specify a rule (AP function) that can guide the agent and help avoid repeated unnecessary trials causing wastage in time. The proposed idea of SAP and the AP function provides only the knowledge of action permissibility in model-free setting (as opposed to learning the complete model of the environment in model-based approach).
>
> C4: could you be more precise about how often the prediction model is updated? What are potential adverse effects if this models keeps overfitting?
>
> R4: We have added the validation curves of the AP prediction models in Appendix. During RL training, we do not need to train the AP predictor for all steps in the whole training period. After the AP predictor is trained for an initial number of steps, we noted that the validation curve saturates to a fairly high accuracy. Thus, we postpone the training of AP predictor, whenever the validation accuracy is above a threshold (delta_acc) and resume its training whenever the validation accuracy falls below delta_acc until it goes above delta_acc again.
>
> C5: There are also limitations in terms of the number of hyperparameters that need be fine-tuned. I would like that the authors include one paragraph discussing in more detail the limitations of their approach.
>
> R5: We have updated the draft with a discussion section to point out the limitations of our method (see Section 6) and also discuss the hyper-parameter tuning in Appendix.

---

### Author Response · Authors · 2018-11-27
**Paper Revised and Uploaded**

We have revised our paper following your comments and addressed your concerns in the revised version. Please consider the revised version as a reference to our responses.

We thank you for reviewing our work and providing valuable feedbacks.

---

### Meta-Review · Area_Chair1 · 2018-12-14
**Interesting idea, but limited applicability**

**Confidence:** 5
**Recommendation:** Reject

**Metareview:**

The paper presents a simple and interesting idea to improve exploration efficiency, using the notion of action permissibility.  Experiments in two problems (lane keeping, and flappy bird) show that exploration can be improved over baselines like DQN and DDPG.  However, action permissibility appears to be very strong domain knowledge that limits the use in complex problems.

Rephrasing one of reviewers, action permissibility essentially implies that some one-step information can be used to rule out suboptimal actions, while a defining challenge in RL is that the agent needs to learn/plan/reason over multiple steps to decide whether an action is suboptimal or not.  Indeed, the two problems in the experiments have such a property that a myopic agent can solve the tasks pretty well.  The paper would be stronger if the AP function can be defined for more common RL benchmarks, with similar benefits demonstrated.